# Structure of the *Helicobacter pylori* Cag type IV secretion system

**Jeong Min Chung[1†], Michael J Sheedlo[2†], Anne M Campbell[3], Neha Sawhney[3], Arwen E Frick-Cheng[2,4], Dana Borden Lacy[2,5]\*, Timothy L Cover[2,3,5]\*, Melanie D Ohi[1,6]\***

[1]Life Sciences Institute, University of Michigan, Ann Arbor, United States; [2]Department of Pathology, Microbiology, and Immunology, Vanderbilt University School of Medicine, Nashville, United States; [3]Department of Medicine, Vanderbilt University School of Medicine, Nashville, United States; [4]Department of Immunology and Microbiology, University of Michigan School of Medicine, Ann Arbor, United States; [5]Veterans Affairs Tennessee Valley Healthcare System, Nashville, United States; [6]Department of Cell and Developmental Biology, University of Michigan School of Medicine, Ann Arbor, United States

**\*For correspondence:**
borden.lacy@vumc.org (DBL);
timothy.l.cover@vumc.org (TLC);
mohi@umich.edu (MDO)

[†]These authors contributed equally to this work

**Competing interests:** The authors declare that no competing interests exist.

**Abstract** Bacterial type IV secretion systems (T4SSs) are molecular machines that can mediate interbacterial DNA transfer through conjugation and delivery of effector molecules into host cells. The *Helicobacter pylori* Cag T4SS translocates CagA, a bacterial oncoprotein, into gastric cells, contributing to gastric cancer pathogenesis. We report the structure of a membrane-spanning Cag T4SS assembly, which we describe as three sub-assemblies: a 14-fold symmetric outer membrane core complex (OMCC), 17-fold symmetric periplasmic ring complex (PRC), and central stalk. Features that differ markedly from those of prototypical T4SSs include an expanded OMCC and unexpected symmetry mismatch between the OMCC and PRC. This structure is one of the largest bacterial secretion system assemblies ever reported and illustrates the remarkable structural diversity that exists among bacterial T4SSs.
DOI: https://doi.org/10.7554/eLife.47644.001

## Introduction

Bacterial pathogens are a threat to global health and have evolved elaborate strategies to infect their hosts. Many effects of bacteria on host cells require the actions of bacterial secretion systems. Bacterial type IV secretion systems (T4SS) are a diverse class of molecular machines that mediate interbacterial DNA transfer through conjugation as well as delivery of effector proteins into host cells. T4SSs are found in a wide range of bacterial species, including many species that cause human disease, such as *Helicobacter pylori*, *Legionella pneumophila*, *Bordetella pertussis*, *Brucella*, and *Bartonella* (*Christie et al., 2014*; *Grohmann et al., 2018*).

T4SSs in Gram-negative bacteria contain a minimum of 12 components (designated VirB1-VirB11 and VirD4 in prototype systems), organized into an outer membrane core complex (OMCC), an inner membrane complex (IMC), and in some species an extracellular pilus (*Christie et al., 2014*; *Grohmann et al., 2018*; *Waksman, 2019*). High-resolution structures have been determined for OMCCs from two minimized T4SSs (*Xanthomonas citri* T4SS and a portion of the OMCC from the pKM101 conjugation system) (*Sgro et al., 2018*; *Chandran et al., 2009*). T4SSs in several bacterial species, including *Helicobacter pylori* and *Legionella pneumophila*, contain additional components, which are not present in the minimized systems (*Grohmann et al., 2018*; *Frick-Cheng et al., 2016*; *Schroeder, 2017*; *Ghosal et al., 2017*; *Chang et al., 2018*; *Chetrit et al., 2018*; *Ghosal et al.,*

**eLife digest** *Helicobacter pylori* is a species of bacterium that can colonize the human stomach, causing changes that greatly increase the risk of ulcers and stomach cancer. Some strains of *H. pylori* produce a protein called CagA, which alters how stomach cells grow and divide. The bacterium injects CagA directly into stomach cells using a syringe-like structure called a type IV secretion system.

Type IV secretion systems are found in many species of bacteria and are involved in a variety of processes, including the exchange of genes between neighboring bacteria. The systems typically have at least 12 components. Previous studies have revealed how the components of some of these systems fit together to form working machines. However, the type IV secretion system that delivers CagA (called the Cag T4SS) contains additional components and it remains unclear how these components are organized in the structure.

A technique called cryo-electron microscopy uses electrons to visualize proteins that have been rapidly frozen so they can be captured and imaged in their natural shape and form. Chung, Sheedlo et al. extracted the Cag T4SS apparatus directly from *H. pylori* and used cryo-electron microscopy to determine its shape to a high level of detail. These images were then used to build a detailed model of the Cag T4SS that included many of its components.

The model shows that the Cag T4SS is larger and more complex than other type IV secretion systems that have been studied previously. Therefore, Chung, Sheedlo et al. propose that the Cag T4SS is specially adapted to work in the stomach.

These findings open the door for future research to define how individual components of the Cag T4SS help to inject CagA into stomach cells. In addition, future research will allow researchers to understand how the type IV secretion systems found in different bacterial species carry out a wide range of roles.

DOI: https://doi.org/10.7554/eLife.47644.002

*2019*; *Hu et al., 2019*). The *H. pylori* Cag T4SS is of particular interest because of its role in translocating CagA (a bacterial oncoprotein) into host cells, an important step in gastric cancer pathogenesis (*Fischer, 2011*; *Backert et al., 2017*). In addition to containing unique components, the *H. pylori* Cag T4SS has an OMCC much larger in size and more intricate than those in minimized systems, and contains a periplasmic sub-complex not seen in minimized T4SSs (*Grohmann et al., 2018*; *Chang et al., 2018*; *Hu et al., 2019*). Here we report the use of single particle cryo-EM to determine the structure of a transmembrane Cag T4SS complex extracted and purified from *H. pylori*. The structure can be divided into three major regions: the OMCC, a periplasmic ring complex (PRC), and a central stalk. The OMCC has a structural organization markedly different from that of T4SS OMCCs in other species, and there is an unexpected symmetry mismatch between the 14-fold-symmetric OMCC and a contiguous 17-fold-symmetric PRC. We propose that the observed structural differences between the Cag T4SS and previously described minimized T4SSs have important functional implications for how T4SSs secrete various kinds of effectors.

## Results

Cag T4SS complexes were purified from *H. pylori* as described (*Frick-Cheng et al., 2016*) and visualized by single particle cryo-EM (*Figure 1* and *Figure 1—figure supplement 1A,B*). The resulting structure is a large mushroom-shaped complex, ~410 Å wide by ~460 Å long, with features that closely match those of the Cag T4SS detected in intact *H. pylori* using cryo-electron tomography (*Chang et al., 2018*; *Hu et al., 2019*) (*Figure 1A,B*). The global resolution of the map is 5.4 Å when no symmetry is imposed, with the highest resolution regions found near the center of the complex (*Figure 1A* and *Figure 1—figure supplement 1C–E*). Although particles adopt a preferred orientation in vitrified ice, both *en face* and side views are present, which allowed for 3D reconstruction (*Figure 1—figure supplement 1B,F*). The map can be divided into three major regions: an intricately organized domed cap that is associated with the outer membrane (OMCC), a hollow ring-like mid-region localized within the periplasm (PRC), and a tapered density that extends from the PRC to the inner membrane (Stalk) (*Figure 1A* and *Video 1*). Blurry density around the PRC, also visible in

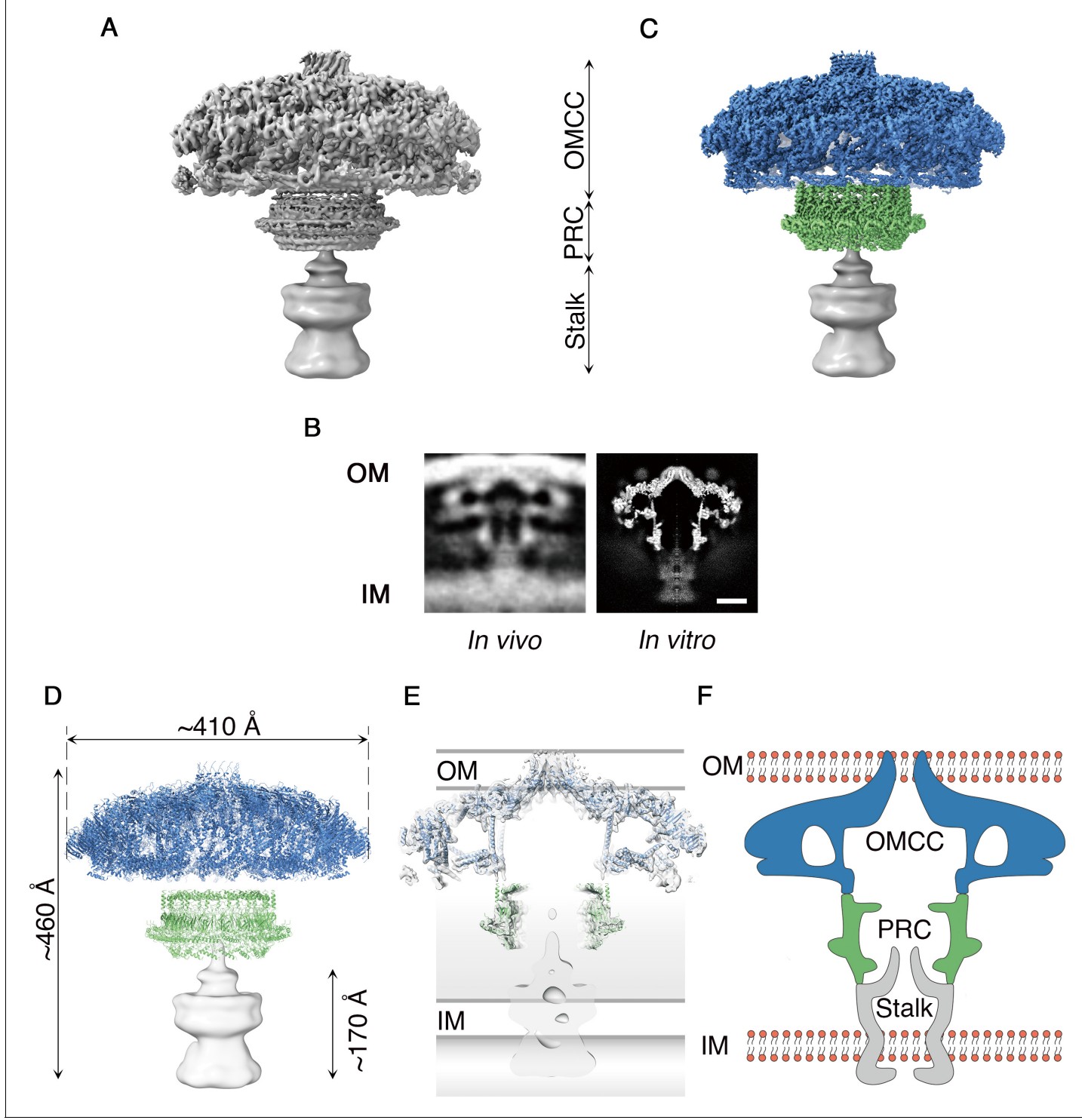

**Figure 1.** Cryo-EM structure of the *H. pylori* Cag T4SS. (**A**) Reconstruction of the *H. pylori* Cag T4SS particles with no applied symmetry at 5.4 Å, showing three parts: the outer membrane core complex (OMCC), periplasmic ring complex (PRC), and Stalk density. The Stalk was not well defined in the structure and has been gaussian filtered to a resolution of 15 Å. (**B**) Comparison of central sections through the longitudinal plane of the T4SS 3D density determined by cryo-ET of intact *H. pylori* (left panel, EMD: 7474) (*Chang et al., 2018*) or by cryo-EM of purified particles (this study) (right panel). OM, Outer membrane, IM, inner membrane. Scale bar, 10 nm **C**. Combined high resolution structure of the *H. pylori* Cag T4SS that includes the 3.8 Å OMCC (blue) with 14-fold symmetry, the 3.5 Å PRC (green) with 17-fold symmetry, and the gaussian filtered Stalk (gray). (**D**) Secondary structure model of the refined OMCC (blue) and PRC (green) with gaussian filtered Stalk (gray). (**E**) Central axial slice view showing how refined structures of the

*Figure 1 continued on next page*

*Figure 1 continued*

OMCC (blue) and PRC (green) fit into the 3D map of the Cag T4SS (light gray). (**F**) Cartoon schematic showing the organization of the *H. pylori* Cag T4SS in relation to the OM and IM. OMCC (blue), PRC (Green) and Stalk (Gray).

DOI: https://doi.org/10.7554/eLife.47644.003

The following figure supplements are available for figure 1:

**Figure supplement 1.** Single particle cryo-EM analysis of *H. pylori* Cag T4SS.

DOI: https://doi.org/10.7554/eLife.47644.004

**Figure supplement 2.** Focused and asymmetric refinement of the *H. pylori* Cag T4SS OMCC.

DOI: https://doi.org/10.7554/eLife.47644.005

cryo-ET images (*Chang et al., 2018*; *Hu et al., 2019*), may represent a dynamic or less-structured portion of the T4SS (*Figure 1B*). Using symmetry, focused refinement, and symmetry expansion, we determined a 3.7 Å resolution map of the OMCC and a 3.5 Å map of the PRC (*Figure 1C*, *Figure 1—figure supplement 2*, *Videos 2* and *3*). The resolution of the OMCC and PRC maps made it possible to trace the secondary structure in these regions (*Figure 1D* and *Videos 2* and *3*). Since only a small number of T4SS particles seen in vitrified ice contained the Stalk density, it was not possible to determine the symmetry or obtain high resolution of this part of the complex. An axial section through the map in *Figure 1D* shows a large cavity running through the Cag T4SS, starting from where the OMCC spans the outer membrane (OM) and extending to the bottom of the PRC (*Figure 1E,F*). The tapered end of the Stalk begins in the PRC channel and continues through the inner membrane (IM). Models developed from cryo-ET analyses of both the *H. pylori* and *L. pneumophila* T4SSs propose a channel in this region of the complex (*Ghosal et al., 2017*; *Chang et al., 2018*; *Chetrit et al., 2018*; *Ghosal et al., 2019*; *Hu et al., 2019*). A central section through the longitudinal plane of 3D density suggests there may be a channel that runs through the Stalk (*Figure 1B,F*). However, due to the low resolution of the Stalk, this channel cannot be clearly visualized in the 3D map.

The OMCC, with 14-fold symmetry, makes up the 'mushroom cap' of the T4SS and is organized into a central and outer ring connected by 'spokes' (*Figure 2*). The outer ring of the OMCC surrounds a central chamber that is ~270 Å wide and tapers to a ~ 35 Å opening at the top (*Figure 2B*). As seen in OMCCs from minimized T4SSs (*Sgro et al., 2018*; *Chandran et al., 2009*), the *H. pylori* Cag T4SS OMCC contains both an outer layer (O-layer) and a thinner inner layer (I-layer) (*Figure 3A*). 'Spokes' and outer rings are visible in both the O-layer and I-layer of the Cag T4SS (*Figure 3B* and *Figure 3—figure supplement 1*), but are not present in OMCCs from minimized T4SSs. The resolution of the cap is high enough to begin mapping the molecular organization of individual OMCC components (*Figure 4A,B*). Previous mass spectrometry and Western blot analyses indicated that the isolated complexes contain CagY, CagX, CagM, CagT, and Cag3 (*Frick-Cheng et al., 2016*). In the map, we can trace and identify portions of CagY (residues 1677–1816

and 1850–1907), CagX (residues 349–510), and CagT (residues 26–269) (*Figure 4C* and *Figure 4—figure supplement 1A–C*, left panels).

CagY forms the crest of the cap-like structure of the OMCC and is comprised of β-sheets intertwining to position two helices per protomer atop the cap (*Figure 4A–C* and *Figure 4—figure supplement 1A*, left panel). We predict that these helices breach the outer membrane, resulting in the formation of a channel (*Figure 4B*). The C-terminal portion of CagX is comprised of two β-sheets preceded by a long helix (*Figure 4C* and *Figure 4—figure supplement 1B*, left panel). This helix extends from the top of the O-layer through the central chamber into the I-layer of the OMCC (*Figure 4B*). Adjacent to CagX, we traced a continuous chain 243 residues long corresponding to CagT (*Figure 4C*). It

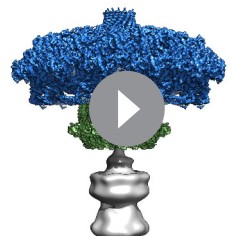

**Video 1.** Movie showing the cryo-EM density maps of the *H. pylori* T4SS.

DOI: https://doi.org/10.7554/eLife.47644.006

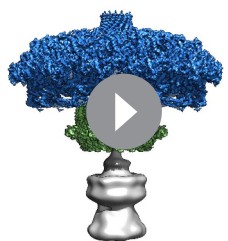

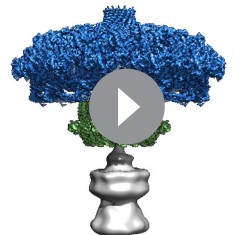

**Video 2.** Movie showing the cryo-EM density of the *H. pylori* T4SS and the molecular map of the OMCC sub-complex.
DOI: https://doi.org/10.7554/eLife.47644.007

**Video 3.** Movie showing the cryo-EM density of the *H. pylori* T4SS and the molecular map of the PRC sub-complex.
DOI: https://doi.org/10.7554/eLife.47644.008

includes a globular subdomain consisting of two β-sheets followed by a long C-terminal extension of three helices that contribute to the spoke and extend toward the edge of the map (*Figure 4B* and *Figure 4—figure supplement 1C*, left panel). A predicted lipidation site within the N-terminal tail of CagT (*Fischer, 2011*; *Akopyants et al., 1998*) was not observed in the cryo-EM map, but the N-terminus of CagT is positioned for this interaction with the outer membrane.

Extending to the edge of the O-layer, we have constructed poly-alanine models of protein(s) that we could not identify (*Figure 4A,B* and *Figure 4—figure supplement 1D*). These components fill the remaining density within the spokes and consist predominantly of a repetitive fold composed of repeating units of β-sheets flanked by helices (*Figure 4A,B* and *Figure 4—figure supplement 1D*). We predict that Cag3 is a component of the OMCC periphery based on a previous study showing that Cag T4SS complexes isolated from a Δ*cag3* mutant strain lacked peripheral components of the OMCC (*Frick-Cheng et al., 2016*; *Hu et al., 2019*), but we were unable to obtain a register. Within the I-layer we have observed at least two distinct bundles of helices of ~200 residues and ~300 residues (*Figure 4A,B* and *Figure 4—figure supplement 1E*). Based on structural studies of the *X. citri* T4SS (*Grohmann et al., 2018*), the I-layer of the Cag T4SS is predicted to contain portions of CagY and CagX, but none of the poly-alanine models allowed us to unambiguously attribute any portion of CagY, CagX, or other T4SS components to this region.

The PRC, with 17-fold symmetry (*Figure 4D,E* and *Figure 4—figure supplement 2*), is a short hollow tube, 90 Å tall and 185 Å wide with 96 Å internal diameter (*Figure 4D,E*), connecting the OMCC and the Stalk regions of the Cag T4SS (*Figure 1A*). Although PRCs have not been detected in structural studies of *E. coli* or *X. citri* T4SSs (*Waksman, 2019*), this region was identified in cryo-ET studies of both *H. pylori* and *L. pneumophila* T4SSs (*Ghosal et al., 2017*; *Chang et al., 2018*; *Chetrit et al., 2018*; *Ghosal et al., 2019*; *Hu et al., 2019*). The protein(s) in this region of the cryo-EM maps were modeled as three distinct polyalanine chains (*Figure 4—figure supplement 1F–H*). This portion of the *H. pylori* Cag T4SS structure is predicted to contain CagM (*Waksman, 2019*; *Chang et al., 2018*), but the three polyalanine chains could not be unambiguously attributed to CagM or other T4SS components. A symmetry mismatch occurs between the OMCC and PRC, going from 14-fold symmetry in the OMCC to 17-fold symmetry in the PRC. While the OMCC, PRC, and Stalk make physical contact in the lower resolution structure with no applied symmetry (*Figure 5*), in the refined structures these connections are lost because of the symmetry mismatch. Analysis of the 5.4 Å structure suggests that the connection between OMCC and PRC may occur by contacts between the 14 α-helices extending from the N-terminus of CagX and the 17 unidentified α-helices extending up from the PRC (*Figure 1A,E*, and *Figure 5*). The tips of these α-helices are presumed to be flexible, since the corresponding densities are not well defined in the higher resolution maps (*Figure 4D*).

 

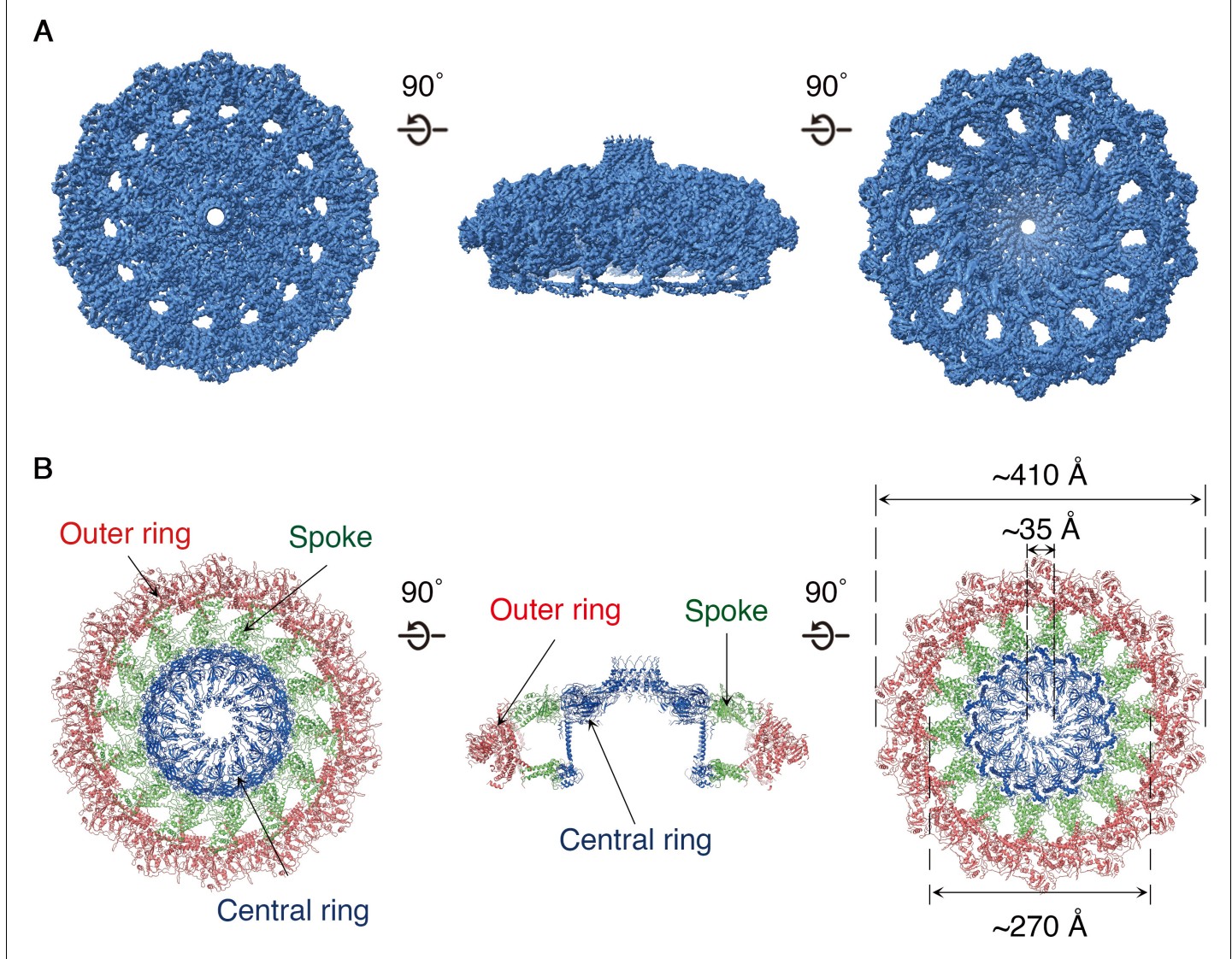

**Figure 2.** Structure of the *H. pylori* Cag T4SS OMCC. (**A**) 3.8 Å resolution cryo-EM density map of the Cag T4SS OMCC rotated 90° around the X-axis. The structure has 14-fold symmetry. Left panel represents view of the OMCC looking in from outer membrane towards the inner membrane, and right panel represents the view looking out from the inner membrane towards the outer membrane. (**B**) Secondary structure model of the OMCC. In the *en face* view on the left (*i.e.* view looking in from the outer membrane), the structure has a central ring (blue) and outer ring (red) connected by 14 spokes (green). The middle panel represents a central slice of the OMCC. The right panel shows the view looking out from the inner membrane.
DOI: https://doi.org/10.7554/eLife.47644.009

## Discussion

Overall, the *H. pylori* OMCC is much larger (410 Å diameter) than other structurally characterized OMCCs (225 Å in *X. citri* and 170 Å in pKM101 conjugation system) (*Sgro et al., 2018*; *Chandran et al., 2009*) (*Figure 6*), and the components are more intertwined. While the inner chamber of the *H. pylori* OMCC is larger than what is seen in the *E. coli* and *X. citri* structures, the dimension of the *H. pylori* outer membrane pore, at 35 Å, is smaller than the *X. citri* outer membrane pore (45 Å) but larger than the *E. coli* outer membrane pore (25 Å) (*Figure 6B*). The dimensions of the *H. pylori* OMCC resemble those of the *L. pneumophila* Dot/Icm T4SS (*Ghosal et al., 2017*; *Chetrit et al., 2018*; *Ghosal et al., 2019*), but there is very little sequence relatedness when comparing components of the *H. pylori* and *L. pneumophila* T4SSs, and the *L. pneumophila* T4SS OMCC has 13-fold symmetry instead of 14-fold symmetry (*Chetrit et al., 2018*; *Ghosal et al., 2019*). The

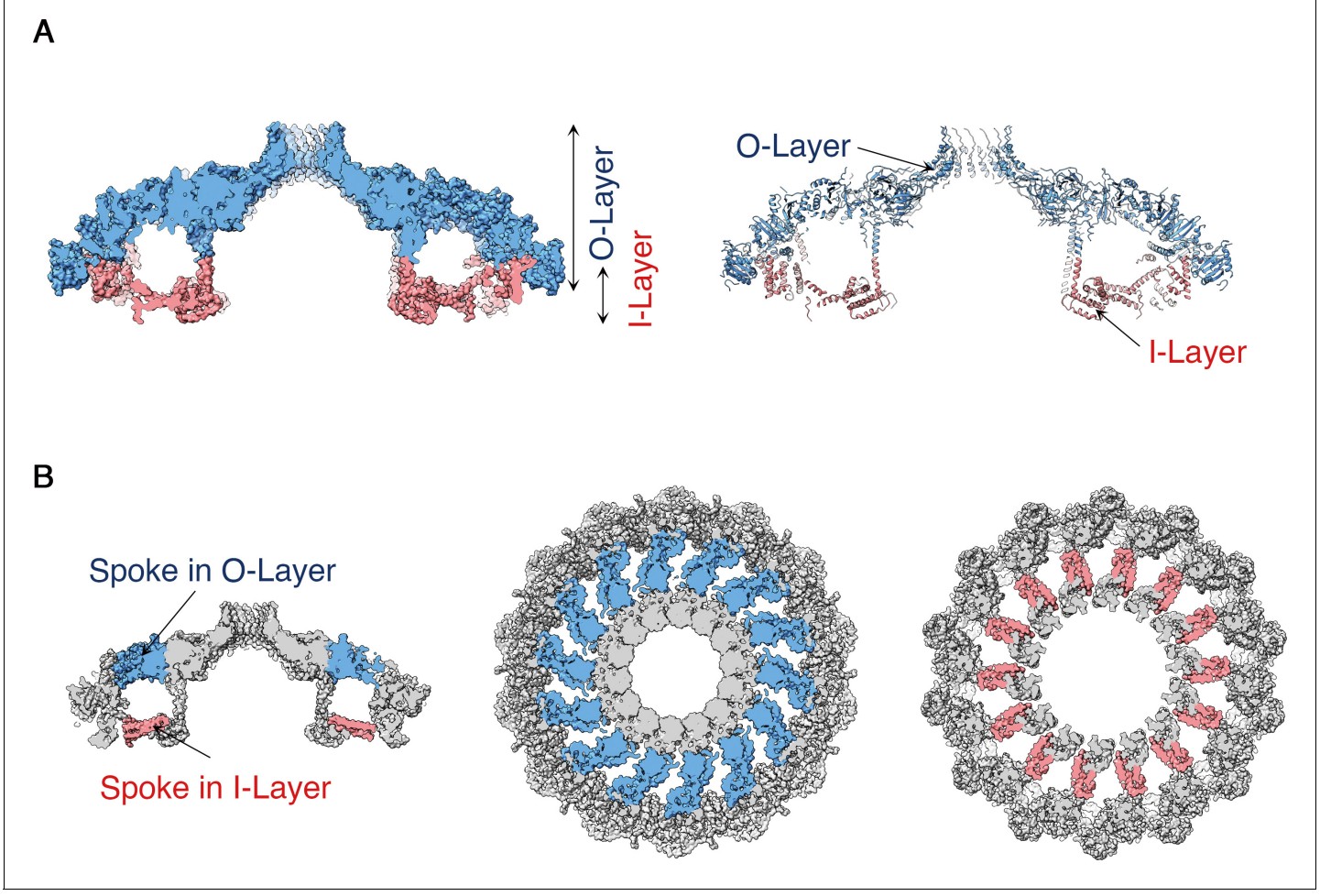

**Figure 3.** Structural features of the *H. pylori* Cag T4SS OMCC I-Layer and O-layer. (**A**) Central axial slice view of the OMCC highlighting the position of the outer-layer (O-layer) and inner-layer (I-layer) colored in blue and red, respectively. Left panel, Surface representation; Right panel, Ribbon representation of secondary structure model. (**B**) Central axial slice view (left) and two cross sections of the OMCC highlighting the position of the spokes in O-layer (middle panel, blue, looking in from the outer membrane) and I-layer (right panel, red, looking out from the inner membrane).

DOI: https://doi.org/10.7554/eLife.47644.010

The following figure supplement is available for figure 3:

**Figure supplement 1.** Spokes in O-layer and I-layer of OMCC.

DOI: https://doi.org/10.7554/eLife.47644.011

C-terminal portion of CagY is structurally similar to VirB10 and TraF homologs (*Figure 6* and *Figure 4—figure supplement 1A*), and the C-terminal portion of CagX is structurally similar to VirB9 and TraO homologs (*Figure 6* and *Figure 4—figure supplement 1B*). Both CagY and CagX are much larger in size than characterized VirB/Tra homologs (*Fischer, 2011*; *Backert et al., 2017*), and correspondingly, CagX has a long α-helix that is absent in the structures of TraO or VirB9 (*4, 5*) (*Figure 6* and *Figure 4—figure supplement 1B*). Consistent with the limited sequence relatedness of CagT to VirB7 homologs (*Fischer, 2011*; *Backert et al., 2017*), there is relatively little structural relatedness when comparing CagT with VirB7 except for an N-terminal portion that contains little secondary structure (*Figure 6* and *Figure 4—figure supplement 1C*).

In summary, these results provide the first high resolution structure of a transmembrane complex from a non-canonical T4SS and provide important new insights into a bacterial T4SS that contributes to the pathogenesis of gastric cancer. The Cag T4SS structure differs markedly from structures of previously described T4SSs, including the presence of an expanded OMCC with components that are more structurally intertwined and an unexpected symmetry mismatch between the OMCC and

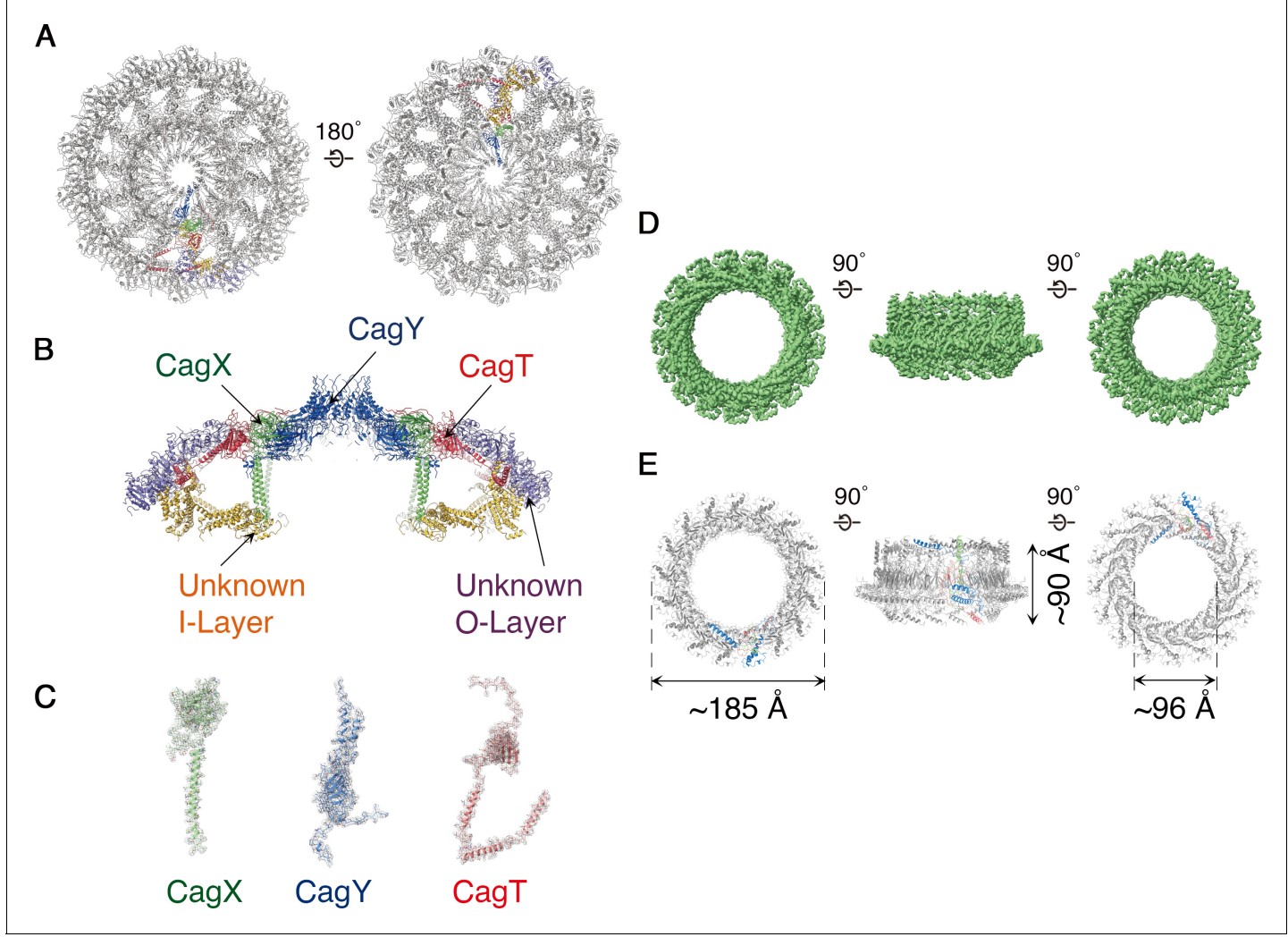

**Figure 4.** Molecular organization of the *H. pylori* Cag T4SS. (**A**) Secondary structure model of the OMCC. The left panel represents view looking in from the outer membrane and the right panel represents the view looking out from the inner membrane. (**A–C**) Blue, CagY; Green, CagX; Red, CagT; Gold, I-layer; and Purple, O-layer. (**B**) Central axial slice view of the secondary structural model of the OMCC highlighting the position of CagY, CagX, CagT, I-layer, and O-layer. (**C**) Cryo-EM densities with built models of CagX, CagY, and CagT. Cryo-EM density map is in gray mesh and model is shown as ribbon diagram. (**D**) 3.5 Å resolution cryo-EM density map of the Cag T4SS PRC rotated 90˚ around the X-axis. The structure has 17-fold symmetry. (**E**) Secondary structure model of the PRC. Chain 1 (67 residues, green), Chain 2 (96 residues, blue) and Chain 3 (136 residues, red). In both **D** and **E** the left panel represents view looking in from outer membrane and right panel represents the view looking out from the inner membrane.

DOI: https://doi.org/10.7554/eLife.47644.012

The following figure supplements are available for figure 4:

**Figure supplement 1.** Structures of T4SS components.
DOI: https://doi.org/10.7554/eLife.47644.013
**Figure supplement 2.** Focused refinement of the *H.pylori* Cag T4SS PRC.
DOI: https://doi.org/10.7554/eLife.47644.014

PRC. We predict that these differences will have important functional implications for the mechanism of CagA translocation.

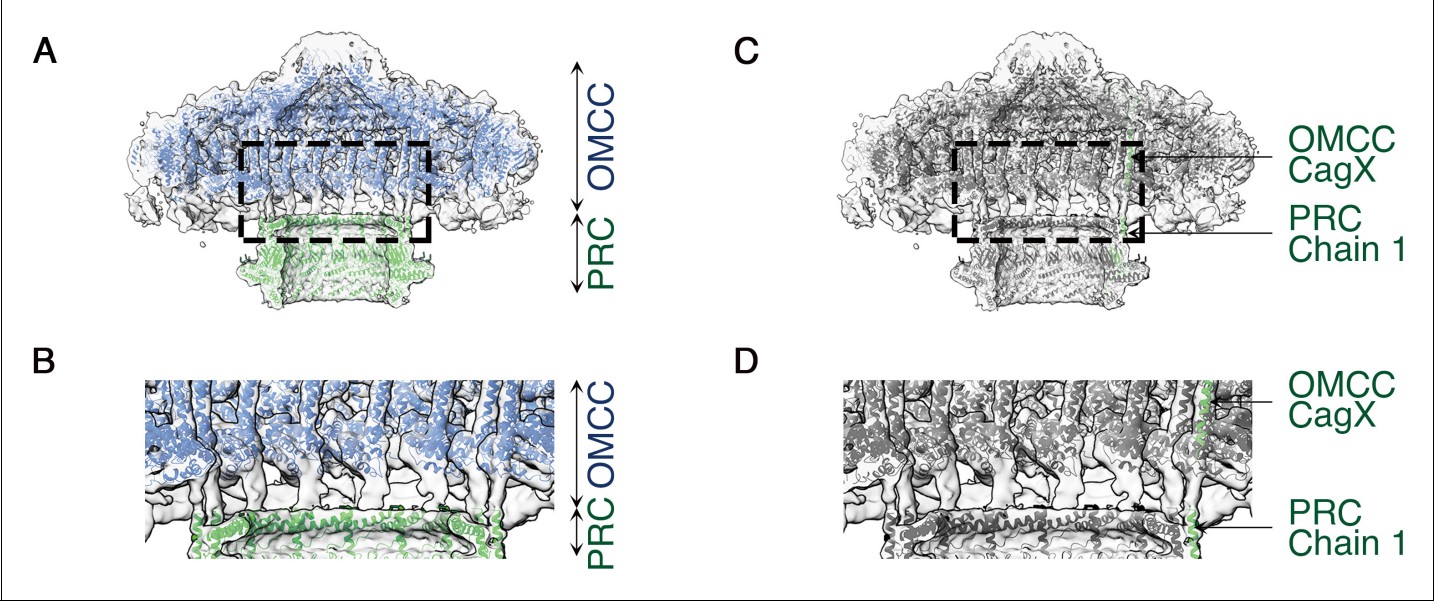

**Figure 5.** C1 symmetry density map showing connection of the OMCC to the PRC. (A) Central axial slice view showing refined structures of the OMCC (blue) and PRC (green) fit into the C1 3D map of the Cag T4SS (light gray). The map is contoured at a low level to show less-ordered features. (B) Zoomed-in view of the region delimited by the dashed line in (a) showing the connection region of OMCC and PRC. (C) Central axial slice view showing refined structures of OMCC CagX (top, green) and PRC Chain 1 (bottom, green) fit into the C1 3D map of Cag T4SS complex (light gray). (D) Zoomed-in view of the region delimited by the dashed line in (c) showing candidates for the connection of OMCC and PRC.

DOI: https://doi.org/10.7554/eLife.47644.015

## Materials and methods

### Purification of the *H. pylori* Cag T4SS core complex

The purification of the Cag T4SS complex was done using a previously described approach (*Frick-Cheng et al., 2016*) with reduced deoxycholate concentration (0.025%).

### EM sample preparation

For cryo-EM, 5 ul of the Cag T4SS sample (as purified) was applied to a glow discharged ultrathin continuous carbon film on Lacey 400 mesh copper grids (TED PELLA). The sample was applied to a grid, incubated for 60 s and vitrified by plunge-freezing in a slurry of liquid ethane using a FEI Vitrobot at 4°C and 100% humidity.

### Cryo-EM data collection

All the images were collected on the Titan Krios electron microscope (Thermo Fisher) equipped with a K2 Summit Direct Electron Detector (Gatan) operated at 300 kV and having a nominal pixel size of 1.64 Å per pixel. Micrographs were acquired using Leginon software (*Suloway et al., 2005*). The total exposure time was 8 s and frames were recorded every 0.2 s, resulting in a total accumulated dose of ~60 e$^-$ Å$^{-2}$ using a defocus range of −0.5 to −3.5 μm.

### Image processing

All the Video frames were first dose-weighted and aligned using Motioncor2 (*Zheng et al., 2017*). The contrast transfer function (CTF) values were determined by Gctf (*Zhang, 2016*). Image processing was carried out using cryoSPARC, RELION 2.1 and 3.0 (*Punjani et al., 2017*; *Scheres, 2012*; *Zivanov et al., 2018*). Using RELION, approximately 25,000 particles were manually picked from 4600 micrographs and extracted using a box size of 510 pixels (836.4 Å) (*Supplementary file 1*). The extracted particles were exported to cryoSPARC and used to generate representative two-dimensional (2D) class averages in both cryoSPARC and RELION, and approximately 24,000 (cryoSPARC) and 23,000 (RELION) particles were kept in good class averages. These

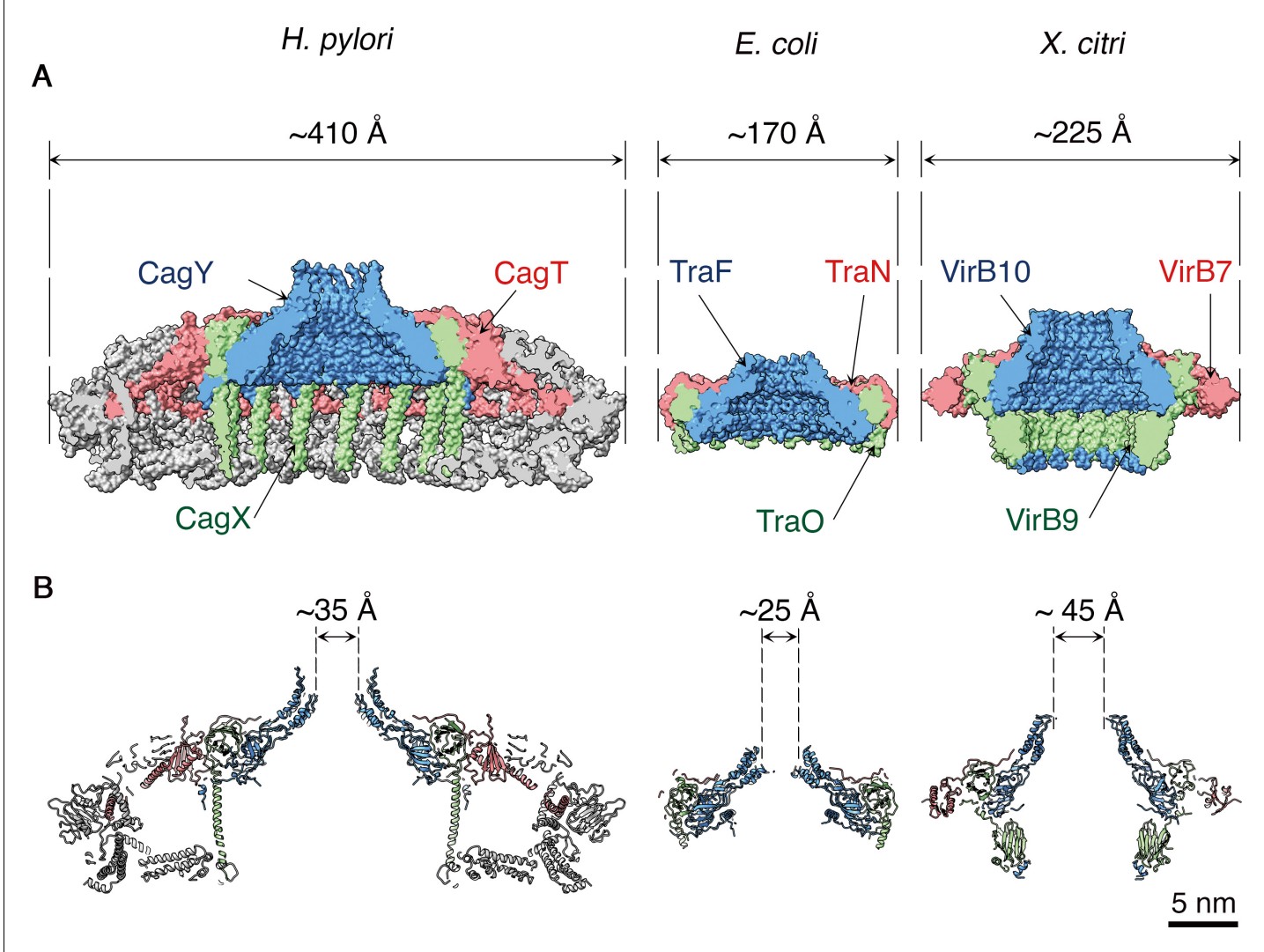

**Figure 6.** Structural comparison of *H. pylori* Cag T4SS OMCC components with *E. coli* and *X. citri* OMCC components. Side views (**A**) and central axial slice views (**B**) of *H. pylori* Cag T4SS (left), *E. coli* pKM101 T4SS (middle; PDB 3JQO [*Chandran et al., 2009*]) and *X. citri* T4SS (right; PDB 6GYB [*Sgro et al., 2018*]). Structural homologs CagY, TraF, and VirB10 are labeled blue. Structural homologs CagX, TraO, and VirB9 are labeled green. Structural homologs CagT, TraN, and VirB7 are labeled red. Scale bar, 5 nm.

DOI: https://doi.org/10.7554/eLife.47644.016

particles were then subjected to 3D classification with a reference-free initial 3D model. The best 3D class (~17,000 in cryoSPARC and ~20,000 particles in RELION) was used as reference for 3D auto-refinement with and without C14 symmetry (low-pass filtered to 60 Å). Finally, a solvent mask and B-factor were applied to improve the overall features and resolution of the 3D maps with and without C14 symmetry, resulting in reconstruction of 3D maps with a global resolution of 4.1 Å and 5.4 Å, respectively (*Supplementary file 1* and *Figure 1—figure supplement 1*). Estimation of per-particle defocus values (CTF-refinement) was applied to the selected particles using RELION. With the CTF-refined particle stack, C14 symmetry-imposed refinement with a soft mask around the OMCC region of the Cag T4SS core complex was done, resulting in a 3.8 Å resolution 3D map that contained improved features (*Figure 1—figure supplement 1* and *Figure 1—figure supplement 2C*). Based on the inspection of the asymmetrically refined 3D map showing 14-fold symmetry, relion_-particle_symmetry_expand was used to enlarge the particle stack (*Zivanov et al., 2018*). After signal subtraction and alignment-free 3D classification of the expanded particles using a soft mask around an asymmetric unit (~290,000 particles), one highly populated 3D class was produced which

contains ~250,000 expanded particles. The 3D class was then subjected to the masked 3D refinement with local angular searches, resulting in reconstruction of the 3D map at 3.7 A° resolution (*Figure 1—figure supplement 2D*). All resolutions were calculated using the gold-standard 0.143 FSC.

To resolve the lower region of the core complex (PRC), signal subtraction for each individual particle containing the PRC signal was used with a soft mask (*Supplementary file 1* and *Figure 4—figure supplement 2A*). The subtracted particles were then subjected to the alignment-free focused 3D classification (five classes). Visual inspection of the 3D classes showed that the PRC has 17-fold symmetry. The best 3D class of the PRC (~20,000 particles) was then subjected to a masked 3D refinement with local angular searches. This process was done using either C1, C12, C14 or C17 symmetry. Only the reconstruction using C17 symmetry improved in resolution, resulting in 3D reconstruction of the PRC at 3.5 A° (*Figure 4—figure supplement 2C–E*). Resolutions were calculated using the gold-standard 0.143 FSC. Image processing steps are summarized in *Supplementary file 1*.

## Model building, refinement, and validation

There were no high-resolution structures for any of the known components of the OMCC except for a crystal structure of a small region in CagX (*Zhang et al., 2017*). Density maps of the OMCC and PRC were of sufficient quality for building de novo models of regions of CagY, CagX, and CagT in COOT (*Figure 4* and *Figure 4—figure supplement 1*), facilitated by the small crystal structure of CagX and homology models of OMCC components from *E. coli* and *X. citri* (*Sgro et al., 2018*; *Chandran et al., 2009*). A homology model for a C-terminal region of CagY was constructed from VirB10 (PDB-3JQO) using the Swiss Model server. The CagX crystal structure (PDB-5H3V) and the CagY model were placed within the asymmetric unit of the electron density map using UCSF Chimera (*Pettersen et al., 2004*). These models were manually adjusted and extended in COOT (*Emsley and Cowtan, 2004*). Density corresponding to CagT was modeled *de novo* in COOT, and the three proteins (CagT, CagX, and CagY) were further refined in real space using Phenix while applying secondary structure restraints (*Adams et al., 2002*). Once the asymmetric unit was constructed, 14-fold symmetry was applied in Phenix, and the entire model of the OMCC was subjected to one more round of refinement. Secondary structure elements were built *de novo* into the remaining density of the OMCC and the PRC asymmetric units in COOT. These models were refined in Phenix as described above, and the entire model was generated by applying 14-fold and 17-fold symmetry to the OMCC and PRC, respectively. The resolution of each individual model was estimated by Fourier Shell correlation against the map used to construct it using the Phenix Cryo-EM Validation tool. Molprobity scores, Clashscores and Ramachandran plots were used to validate the models that were constructed (*Chen et al., 2010*; *Afonine et al., 2018*) (*Supplementary file 2*). *Supplementary file 3* shows the FSCs of the half maps against the refined model agree with each other, suggesting that the models are not over-refined. Programs used for structure determination and refinement were accessed through SBGrid (*Morin et al., 2013*). Structures were rendered using Chimera and ChimeraX (*Pettersen et al., 2004*; *Goddard et al., 2018*).

## Data availability

The cryo-EM volumes have been deposited in the Electron Microscopy Data Bank under accession codes EMD-20023 (T4SS C1 reconstruction), EMD-20020 (Focused OMCC Reconstruction), EMD-20022 (OMCC Asymmetric Reconstruction), EMD-20021 (Focused PRC Reconstruction). Map coordinates have been deposited in the Protein Data Bank under accession numbers 6OEE (CagT), 6OEG (CagX), and 6ODI (CagY), 6OEF (O-layer), 6OEH (I-Layer), and 6ODJ (PRC).

## Acknowledgements

We thank M Cianfrocco for cloud computing advice and M Su for imaging and data processing advice. We thank the Cianfrocco, Cover, Lacy, and Ohi labs for helpful discussions. We acknowledge the use of the U-M LSI cryo-EM facility, managed by M Su and A Bondy, and U-M LSI IT support. We also thank B Carragher and C Potter for support from the National Resource for Automated Molecular Microscopy located at the New York Structural Biology Center. This work was supported by NIH AI118932, CA116087, GM103310, and the Department of Veterans Affairs 1I01BX004447.

# Additional information

## Funding

| Funder | Grant reference number | Author |
|---|---|---|
| National Institute of Allergy and Infectious Diseases | AI118932 | Timothy L Cover<br>Melanie D Ohi |
| National Cancer Institute | CA116087 | Timothy L Cover |
| U.S. Department of Veterans Affairs | 1I01BX004447 | Timothy L Cover |
| National Institute of General Medical Sciences | GM103310 | Melanie D Ohi |

The funders had no role in study design, data collection and interpretation, or the decision to submit the work for publication.

## Author contributions

Jeong Min Chung, Conceptualization, Formal analysis, Validation, Investigation, Visualization, Methodology, Writing—original draft, Writing—review and editing; Michael J Sheedlo, Data curation, Formal analysis, Validation, Visualization, Writing—original draft, Writing—review and editing; Anne M Campbell, Investigation, Methodology; Neha Sawhney, Methodology, Writing—review and editing; Arwen E Frick-Cheng, Investigation, Methodology, Writing—review and editing; Dana Borden Lacy, Resources, Formal analysis, Supervision, Validation, Investigation, Visualization, Writing—original draft, Writing—review and editing; Timothy L Cover, Conceptualization, Resources, Formal analysis, Supervision, Funding acquisition, Validation, Investigation, Visualization, Writing—original draft, Writing—review and editing; Melanie D Ohi, Conceptualization, Resources, Formal analysis, Supervision, Funding acquisition, Validation, Investigation, Visualization, Methodology, Writing—original draft, Writing—review and editing

## Author ORCIDs

Jeong Min Chung (iD) https://orcid.org/0000-0002-4285-8764
Michael J Sheedlo (iD) https://orcid.org/0000-0002-3185-1727
Neha Sawhney (iD) http://orcid.org/0000-0002-4943-1018
Timothy L Cover (iD) https://orcid.org/0000-0001-8503-002X
Melanie D Ohi (iD) https://orcid.org/0000-0003-1750-4793

## Decision letter and Author response

Decision letter https://doi.org/10.7554/eLife.47644.042
Author response https://doi.org/10.7554/eLife.47644.043

# Additional files

## Supplementary files

• Supplementary file 1. Flow chart of cryo-EM processing steps. ~25,000 Cag T4SS particles were manually picked in RELION and analyzed by two different image processing software packages (CryoSPARC and RELION). The processing steps done in CryoSPARC are on a gray background and the processing steps done using RELION are on a tan background. The particles were exported into cryoSPARC for 2D alignment and *ab initio* 3D classification. The best class was chosen for further refinement, without symmetry (C1). The 5.4 Å 3D model with no symmetry applied was filtered to 60 Å resolution and used as an initial model for 3D structure determination in RELION using 3D refinement without applied symmetry. Focused 3D classification (without alignment) was used to determine higher resolution maps of the OMCC (with 14-fold symmetry) and the PRC (with 17-fold symmetry). The maps of the OMCC and PRC were further refined using focused 3D refinement (with local refinement), resulting in 3D maps of the OMCC (14-fold symmetry) and the PRC (17-fold

symmetry) at 3.8 Å and 3.5 Å, respectively. To improve the resolution of the OMCC, symmetry expansion was applied, resulting in the 3D reconstruction at 3.7 Å resolution.

DOI: https://doi.org/10.7554/eLife.47644.017

• Supplementary file 2. Summary of Data collection, model refinement, and model validation. (A) Information about the cryo-EM data collection and data sets. (B) Information about model and model validation.

DOI: https://doi.org/10.7554/eLife.47644.018

• Supplementary file 3. FSCs of the half maps against the refined models agree with each other, suggesting that the models are not over-refined. A-F. FSC of the half map against the refined model of CagT (A), CagX (B), CagY (C), O-layer (D), I-layer (E), and PRC (F).

DOI: https://doi.org/10.7554/eLife.47644.019

• Transparent reporting form

DOI: https://doi.org/10.7554/eLife.47644.020

### Data availability

The cryo-EM volumes have been deposited in the Electron Microscopy Data Bank under accession codes EMD-20023 (T4SS C1 reconstruction), EMD-20020 (Focused OMCC Reconstruction), EMD-20022 (OMCC Asymmetric Reconstruction), EMD-20021 (Focused PRC Reconstruction). Map coordinates have been deposited in the Protein Data Bank under accession numbers 6OEE (CagT), 6OEG (CagX), and 6ODI (CagY), 6OEF (O-layer), 6OEH (I-Layer), and 6ODJ (PRC).

The following datasets were generated:

| Author(s) | Year | Dataset title | Dataset URL | Database and Identifier |
|---|---|---|---|---|
| Chung JM, Sheedlo MJ, Campbell AM, Sawhney N, Frick-Cheng AE, Lacy DB, Cover TL, Ohi MD | 2019 | T4SS C1 reconstruction | https://www.ebi.ac.uk/pdbe/entry/emdb/EMD-20023 | Electron Microscopy Data Bank, EMD-20023 |
| Chung JM, Sheedlo MJ, Campbell AM, Sawhney N, Frick-Cheng AE, Lacy DB, Cover TL, Ohi MD | 2019 | Focused OMCC Reconstruction | https://www.ebi.ac.uk/pdbe/entry/emdb/EMD-20020 | Electron Microscopy Data Bank, EMD-20020 |
| Chung JM, Sheedlo MJ, Campbell AM, Sawhney N, Frick-Cheng AE, Lacy DB, Cover TL, Ohi MD | 2019 | OMCC Asymmetric Reconstruction | https://www.ebi.ac.uk/pdbe/entry/emdb/EMD-20022 | Electron Microscopy Data Bank, EMD-20022 |
| Chung JM, Sheedlo MJ, Campbell AM, Sawhney N, Frick-Cheng AE, Lacy DB, Cover TL, Ohi MD | 2019 | Focused PRC Reconstruction | https://www.ebi.ac.uk/pdbe/entry/emdb/EMD-20021 | Electron Microscopy Data Bank, EMD-20021 |
| Chung JM, Sheedlo MJ, Campbell AM, Sawhney N, Frick-Cheng AE, Lacy DB, Cover TL, Ohi MD | 2019 | CagT | https://www.rcsb.org/structure/6OEE | Protein Data Bank, 6OEE |
| Chung JM, Sheedlo MJ, Campbell AM, Sawhney N, Frick-Cheng AE, Lacy DB, Cover TL, Ohi MD | 2019 | CagX | https://www.rcsb.org/structure/6OEG | Protein Data Bank, 6OEG |
| Chung JM, Sheedlo MJ, Campbell AM, | 2019 | CagY | https://www.rcsb.org/structure/6ODI | Protein Data Bank, 6ODI |

| | | | | |
|---|---|---|---|---|
| Sawhney N, Frick-Cheng AE, Lacy DB, Cover TL, Ohi MD | | | | |
| Chung JM, Sheedlo MJ, Campbell AM, Sawhney N, Frick-Cheng AE, Lacy DB, Cover TL, Ohi MD | 2019 | O-layer | https://www.rcsb.org/structure/6OEF | Protein Data Bank, 6OEF |
| Chung JM, Sheedlo MJ, Campbell AM, Sawhney N, Frick-Cheng AE, Lacy DB, Cover TL, Ohi MD | 2019 | I-layer | https://www.rcsb.org/structure/6OEH | Protein Data Bank, 6OEH |
| Chung JM, Sheedlo MJ, Campbell AM, Sawhney N, Frick-Cheng AE, Lacy DB, Cover TL, Ohi MD | 2019 | PRC | https://www.rcsb.org/structure/6ODJ | Protein Data Bank, 6ODJ |

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
