## [Decision Letter]

Thank you for submitting your article "Structure of the *Helicobacter pylori* Cag Type IV Secretion System" for consideration as a short report by *eLife*. Your article has been reviewed by three peer reviewers, one of whom is a member of our Board of Reviewing Editors, and the evaluation has been overseen by Gisela Storz as the Senior Editor. The reviewers have opted to remain anonymous.

Following discussion among the reviewers, we would like to invite you to prepare a revised submission. There was strong support for publishing your paper. It is the first near-atomic structure of the transmembrane subcomplex from a non-canonical T4SS and this allows for exciting comparisons to previously determined structures.

Summary:

In this study, Chung et al. used cryo-EM to reveal a near-atomic structure of the membrane-spanning *Helicobacter pylori* Cag T4SS complex. The authors, in their earlier study, used negative staining electron microscopy to report the overall shape of the *Helicobacter pylori* Cag T4SS transmembrane complex and revealed that this complex is comprised of five *H. pylori* proteins, CagM, CagT, Cag3, CagX, and CagY. This is an extension of their previous work. Here, the authors used cryoEM to resolve the membrane-spanning complex at 3-7 Å resolution. The authors showed that the transmembrane cag T4SS has three sub-complexes: a 14-fold symmetric outer membrane core complex (OMCC), 17-fold symmetric periplasmic complex (PRC), and connecting the PRC to the inner membrane is a stalk domain. They discovered an unexpected mismatch in symmetry between the OMCC and the PRC. Using previously reported structures and where possible, by using homology modeling the authors were able to model part of the OMCC. For rest of the OMCC and PRC map, the authors modeled with polyalanine chains. The stalk domain did not resolve well. This is the first near atomic structure of the OMCC/PRC transmembrane subcomplex from a noncanonical T4SS (canonical conjugative T4SS are 12-component minimized systems). This is a really exciting and dramatic advance in resolution in our structural understanding of the Cag T4SS. It is particularly interesting to compare this structure to the previously-determined structures of related systems. The new structural info is wonderful, but surprisingly the paper contains very little new biological insight.

Essential revisions:

1) Some of the structural figures do not clearly depict what the authors are describing, and should be clarified before publication.

a) Figure 2B does not successfully show the existence of a chamber – I would recommend reconfiguring the third (right-most) image to a central slice akin to Figure 1E or Figure 3A.

b) Figure 4A could be removed as it is confusing and unclear, and does not show anything that Figure 4B does not. However, I also find Figure 4B to be overly dense and unclear. The colours of CagY and the unknown O-layer are too similar. Perhaps the authors could present Figure 4B as in Figure 4E, in which only one of the symmetric units is shown in colour with the others being gray. As in Figure 4E, two orthogonal views would help. I think this is an important point, because the overall organization of the complex is currently hard to understand even after close examination.

c) Similarly, Figure 6A is confusing and unnecessary because the colours are inconsistent with Figure 6B, and most of the coloured area in the *H. pylori* structure are concealed. Figure 6B is much clearer, and effectively illustrates the comparisons made in the text by on its own, as is.

2) The authors can only build part of the OMCC and PRC and rest of the density it fitted with polyalanine chains. The authors mention these additional protein densities as 'unknown' protein densities. Since from their earlier paper it is already known which all proteins form the OMCC and PRC and they have purified particles, the authors in their future work might wish to perform cross-linking mass-spectrometry and identify these additional densities in the OMCC and in the PRC. This will reveal the exact constituents of the OMCC and PRC and provide interesting new insights.

3) Flowchart in the Supplementary File 1 needs major revision as is the flowchart is difficult to follow. Please consider following:

a) The wording '2D filtered' with cryoSPARC gives a wrong impression. This could be something like selected 'good' particles?

b) You could make another box just below (~24,000 particles ‘2D filtered’) and write ‘ab initio model generation’ then go ‘C1 symmetry’.

c) After 3D refinement (in cryoSPARC), when you are showing, C14 symmetrization, show only the OMCC, showing the PRC and providing 4.1 Å resolution is confusing.

d) In Relion, after 23,000 particles/2D filtered – mention what model was used.

e) Spelling mistake '3D classification'.

f) 'after particle CTF correction', replace: 'after per-particle" CTF refinement'.

g) Focused 3D classification: PRC+Stalk 14 fold symmetry, not sure what this is showing.

h) Focused 3D refinement: C14 symmetry: only show the OMCC, C17 symmetry only show the PRC at their appropriate levels and show a composite. The stalk is much lower resolution.

i) The OMCC map is 3.7 Å or 3.8 Å?

4) The membrane-spanning complex is constituted of the OMCC, PRC and the stalk. In the conjugative T4SS, a similar stalk density was reported by Low et al. However, two recent in situ studies by Chetrit et al. (2018) and Ghosal et al. (2017) have reported that there this a 'channel' below the PRC in the Legionella T4SS. Since the *Legionella* T4SS system matches with the Cag system in terms of number of components, shape and appearance, it is likely that the Cag system also has a channel in situ and it collapsed during detergent purification to form the stalk density. The authors should discuss this.

---

## [Author Response]

Essential revisions:1) Some of the structural figures do not clearly depict what the authors are describing, and should be clarified before publication.a) Figure 2B does not successfully show the existence of a chamber – I would recommend reconfiguring the third (right-most) image to a central slice akin to Figure 1E or Figure 3A.b) Figure 4A could be removed as it is confusing and unclear, and does not show anything that Figure 4B does not. However, I also find Figure 4B to be overly dense and unclear. The colours of CagY and the unknown O-layer are too similar. Perhaps the authors could present Figure 4B as in Figure 4E, in which only one of the symmetric units is shown in colour with the others being gray. As in Figure 4E, two orthogonal views would help. I think this is an important point, because the overall organization of the complex is currently hard to understand even after close examination.c) Similarly, Figure 6A is confusing and unnecessary because the colours are inconsistent with Figure 6B, and most of the coloured area in the H. pylori structure are concealed. Figure 6B is much clearer, and effectively illustrates the comparisons made in the text by on its own, as is.

We thank the reviewers for providing constructive suggestions for how to improve the way we present the *H. pylori* T4SS structure. In response, we have made the following changes:

i) Figure 2B is now shown as a central section through the longitudinal plane of 3D density.

ii) Figure 4A has been removed and replaced with two views of the OMCC with single copies of CagX, CagY, and CagT colored. Figure 4B now shows only a central slice of the OMCC.

iii) Figure 6A has been removed and Figure 6B is now Figure 6A. We have also included images of a central section through the longitudinal plane of 3D density of the *H. pylori, E. coli*, and *X. citri* OMCCs to make it easier to visualize and compare the channels created by CagY, TraF, and VirB10, respectively. This is now Figure 6B.

We have also now included videos, Video 1-3, of the *H. pylori* T4SS and the OMCC and PRC sub-complexes that should also help readers more clearly see the organization of this complex.

2) The authors can only build part of the OMCC and PRC and rest of the density it fitted with polyalanine chains. The authors mention these additional protein densities as 'unknown' protein densities. Since from their earlier paper it is already known which all proteins form the OMCC and PRC and they have purified particles, the authors in their future work might wish to perform cross-linking mass-spectrometry and identify these additional densities in the OMCC and in the PRC. This will reveal the exact constituents of the OMCC and PRC and provide interesting new insights.

We thank the reviewers for this suggestion. We agree that crosslinking followed by mass spectrometry analysis will be an important avenue to pursue in our future work, and we agree that such experiments have the potential to provide important information about the organization of the T4SS, especially the region of the cryo-EM map that we could not assign to specific components. We anticipate that getting the final results from this type of analysis will be challenging because of the low protein concentration of the purified complexes coupled with the sample aggregation that we continue to fight against. For these reasons, the execution and analysis of this experiment would likely take substantially longer than two months to complete. Therefore, the revised manuscript does not contain new data based on crosslinking and mass spectrometry analysis.

3) Flowchart in the Supplementary File 1 needs major revision as is the flowchart is difficult to follow. Please consider following:a) The wording '2D filtered' with cryoSPARC gives a wrong impression. This could be something like selected 'good' particles?b) You could make another box just below (~24,000 particles ‘2D filtered’) and write ‘ab initio model generation’ then go ‘C1 symmetry’.c) After 3D refinement (in cryoSPARC), when you are showing, C14 symmetrization, show only the OMCC, showing the PRC and providing 4.1 Å resolution is confusing.d) In Relion, after 23,000 particles/2D filtered – mention what model was used.e) Spelling mistake '3D classification'.f) 'after particle CTF correction', replace: 'after per-particle" CTF refinement'.g) Focused 3D classification: PRC+Stalk 14 fold symmetry, not sure what this is showing.h) Focused 3D refinement: C14 symmetry: only show the OMCC, C17 symmetry only show the PRC at their appropriate levels and show a composite. The stalk is much lower resolution.i) The OMCC map is 3.7 Å or 3.8 Å?

Using the reviewer suggestions, we have now revised Supplementary file 1 so that it will be easier for readers to follow.

4) The membrane-spanning complex is constituted of the OMCC, PRC and the stalk. In the conjugative T4SS, a similar stalk density was reported by Low et al. However, two recent in situ studies by Chetrit et al. (2018) and Ghosal et al. (2017) have reported that there this a 'channel' below the PRC in the Legionella T4SS. Since the Legionella T4SS system matches with the Cag system in terms of number of components, shape and appearance, it is likely that the Cag system also has a channel in situ and it collapsed during detergent purification to form the stalk density. The authors should discuss this.

We agree with the reviewers that it is likely that the T4SS stalk has a ‘channel’ that runs from the cytoplasmic side of the inner membrane to the top of the stalk. This has also been proposed in models developed from cryo-ET analysis of both the *H. pylori* and *L. pneumophila* T4SSs. We have now added the following sentence to the manuscript:

“Models developed from cryo-ET analysis of both the *H. pylori* and *L. pneumophila* T4SSs propose a channel in this region of the complex (Chang et al., 2018; Ghosal et al., 2017). A central section through the longitudinal plane of 3D density suggests there may be a channel that runs through the Stalk (Figure 1B, F). However, due to the low resolution of the Stalk this channel cannot be clearly visualized in the 3D map.”